# Pseudodoping of a metallic two-dimensional material by the supporting substrate

Bin Shao[1,2], Andreas Eich[3], Charlotte Sanders[4], Arlette S. Ngankeu[5], Marco Bianchi[5], Philip Hofmann[5], Alexander A. Khajetoorians[3] & Tim O. Wehling[1,2]

Charge transfers resulting from weak bondings between two-dimensional materials and the supporting substrates are often tacitly associated with their work function differences. In this context, two-dimensional materials could be normally doped at relatively low levels. Here, we demonstrate how even weak hybridization with substrates can lead to an apparent heavy doping, using the example of monolayer 1H-TaS$_2$ grown on Au(111). Ab-initio calculations show that sizable changes in Fermi areas can arise, while the transferred charge between substrate and two-dimensional material is much smaller than the variation of Fermi areas suggests. This mechanism, which we refer to as pseudodoping, is associated with non-linear energy-dependent shifts of electronic spectra, which our scanning tunneling spectroscopy experiments reveal for clean and defective TaS$_2$ monolayer on Au(111). The influence of pseudodoping on the formation of many-body states in two-dimensional metallic materials is analyzed, shedding light on utilizing pseudodoping to control electronic phase diagrams.

---

[1] Institut für Theoretische Physik, Universität Bremen, Otto-Hahn-Allee 1, 28359 Bremen, Germany. [2] Bremen Center for Computational Materials Science, Universität Bremen, Am Fallturm 1a, 28359 Bremen, Germany. [3] Institute for Molecules and Materials, Radboud University, 6525 AJ Nijmegen, The Netherlands. [4] Central Laser Facility, STFC Rutherford Appleton Laboratory, Harwell, Didcot OX11 0QX, UK. [5] Department of Physics and Astronomy, Interdisciplinary Nanoscience Center (iNANO), Aarhus University, 8000 Aarhus C, Denmark. Correspondence and requests for materials should be addressed to B.S. (email: bin.shao@uni-bremen.de) or to T.O.W. (email: wehling@itp.uni-bremen.de)

The family of two-dimensional (2d) materials has been expanding from graphene type materials to compounds such as 2d oxides, chalcogenides and halides[1–3]. While most of the initial research concentrated on graphene and 2d semiconductors, monolayers of metallic 2d materials, particularly transition metal monochcalcogenides and dichcalcogenides such as FeX (X = Se, Te)[4–7] and MX$_2$ (M = V, Nb, Ta; X = S, Se)[8–12], can now be synthesized, processed and even integrated into van der Waals heterostructures[2]. A central reason for the growing interest in metallic 2d systems is that they host highly intriguing many-body states including competing superconducting, nematic, magnetic, excitonic, and charge ordered phases[4–11].

Generally, the electronic phase diagrams of these materials depend on carrier concentrations. Hence, one particularly important way of controlling many-electron phenomena in 2d materials is doping. It allows to switch, for instance, between insulating, charge-ordered, spin-ordered, or superconducting states of a material[9,13–16]. There is however a problem in many systems: changes to electronic states (e.g., switching the prototypical material of 1T-TaS$_2$ from a Mott insulator to a superconductor[9]) often require electron or hole doping on the order of a few 10% of an electron or hole per unit cell[9,14–16]. This translates into carrier concentrations more than or about $10^{14}$ cm$^{-2}$ which are out of reach for gating in standard field effect transistor geometries but require ionic liquids or chemical means like atom substitution, intercalation etc. Doping at this level is potentially related to severe chemical changes of the material and substantial disorder[17].

Here, we discuss an alternative doping mechanism for metallic 2d materials. This mechanism, which we refer to as pseudodoping, is associated with considerable changes in apparent Fermi areas but much smaller actual charge transfer between substrate and 2d material. The reason why a changed Fermi area does not necessarily imply charge transfer is that the Fermi contour is made up from hybridized states of the 2d material and the substrate. We illustrate that pseudodoping of more than 10% of an electron or hole per unit cell is possible using the example of monolayer 1H-TaS$_2$ on Au(111), which has been recently studied by photoemission spectroscopy[12]. Pseudodoping induced shifts of electronic states are often non-linearly energy-dependent such that they can affect states below and above the Fermi energy differently. In this way, pseudodoping gives a unified explanation

of photoemission spectroscopy experiments[12] and scanning tunneling spectra of TaS$_2$ monolayers on Au(111). Afterwards the generic pseudodoping mechanism is explained within a simple two-band model, which we also use to discuss the influence of pseudodoping on the emergence of charge, magnetic, or superconducting order.

## Results

**Pseudodoping of TaS$_2$ on Au(111).** To study the interaction of 1H-TaS$_2$ monolayer and the Au substrate we performed density functional theory (DFT) calculations and scanning tunneling spectroscopy experiments. For the DFT simulations (see Methods), we constructed a $\sqrt{3} \times \sqrt{3}$ R30° TaS$_2$ supercell on a 2 × 2 supercell of the Au(111) surface, where the Au surface has been laterally compressed by 0.5% to have a commensurate structure. This supercell and rotation are different from what we observe experimentally, but are selected for computational feasibility. We have also examined other geometries (including the experimentally observed one) and find very similar results (see Supplementary Figs. 2 and 3). For the $\sqrt{3} \times \sqrt{3}$ R30° TaS$_2$ supercell on a 2 × 2 supercell of the Au(111) surface, we have considered three possible adsorption geometries all of which yield very similar electronic structures (see Supplementary Fig. 1). In the following we discuss the structure with the lowest energy (see Fig. 1a, b). The adsorption height of TaS$_2$ above the Au(111) surface as denoted here by the vertical distance $h$ between the lowest S atoms and the upmost Au atoms has been optimized yielding the equilibrium distance $h_{eq} = 2.37$ Å, which is indicative of physisorptive coupling between the TaS$_2$ and the substrate.

The influence of the coupling between the TaS$_2$ and the substrate on the electronic structure can be inferred from band structures in Fig. 1c, where the TaS$_2$ layer approaches the Au(111) surface from $h_\infty$, i.e., the limit of a clean Au(111) substrate and a free-standing TaS$_2$ monolayer, via $h = h_{eq} + 2$ Å to $h = h_{eq}$. Upfolded band structures[18,19] of the supercell to the Brillouin zone (BZ) of the TaS$_2$ primitive cell are shown, where we highlight states with Ta-$d$ character as blue dots. It can be seen in Fig. 1c that the Ta-$d$ spectral weight of the adsorbed TaS$_2$ at $h_{eq} + 2$ Å and $h_{eq}$ roughly follows the energy dispersion of the free-standing TaS$_2$ $d$-bands for most parts of the BZ path, particularly below the Fermi level. The $p_z$ spectral weight from the

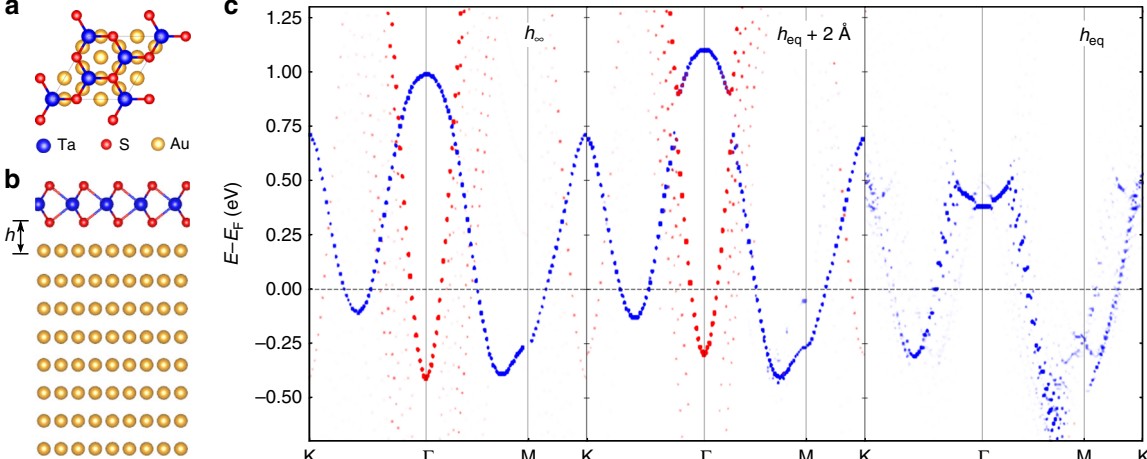

**Fig. 1** Pseudodoping of TaS$_2$ on Au(111). Supercell of monolayer 1H-TaS$_2$ on Au(111), **a** top view and **b** side view. The adsorption height $h$ refers to the height of the lowest S atoms above the uppermost Au atoms. **c** Projection of upfolded band structures on Ta-$d$ orbitals (blue dots) and upmost Au-$p_z$ orbitals (red dots) as a function of adsorption height: $h_\infty$, $h_{eq} + 2$ Å, and $h_{eq}$. The $h_\infty$ case consists of two separated systems: a clean Au(111) substrate and a free-standing 1H-TaS$_2$ monolayer

upmost Au atoms (red dots) results from a Shockley-type surface state with parabolic dispersion. However, there are also some prominent changes in the electronic dispersion of system, as TaS$_2$ gradually approaches its substrate:

First, the Au-$p_z$ derived band at the $\Gamma$ point moves from $-0.4$ eV below the Fermi energy ($E_F$) to more than 1.25 eV above the $E_F$ and thus gets depopulated for a decreasing adsorption height. Such energy shifts of surface states are quite typical for noble metal substrates[20,21]. In contrast, the Ta-$d$ derived band is lowered in energy by about 0.2 eV in the large parts of the BZ path for $h = h_{eq}$ when compared to the free-standing case. This shift applies in particular to the Ta-$d$ states at and below the Fermi level and translates via the density of states (DOS) $\rho(E_F) \approx 2$ eV$^{-1}$ into an apparent doping on the order of 0.4 electrons per TaS$_2$ unit cell. This amount of doping appears unusual given that TaS$_2$ is merely physisorbed on Au(111) but it is in agreement with the experiments: our calculated band structures at $h_{eq}$ (Fig. 1c) well reproduce the occupied part of the energy dispersion measured by angle-resolved photoemission spectroscopy (ARPES) in ref. [12]. Summarizing, DFT and ARPES yield an apparent doping of more than or about 0.3 electrons per unit cell but the physical mechanism behind it is so far unclear.

A charge transfer $\Delta N$ (given in electrons per unit cell) over an effective distance $d$ is associated with an electrostatic potential difference $\Delta V = \alpha d \Delta N$, where $\alpha = e^2/\varepsilon_0 A = 18.6$ eV Å$^{-1}$ and $A = 9.7$ Å$^2$ is the area of the TaS$_2$ unit cell. Even if we assume that the effective distance between the charges in the 2d monolayer and the substrate is $d = 1$ Å, i.e., much smaller than the typical distances of ~4 Å from the center of the TaS$_2$ layer to the upmost substrate atoms, a charge transfer of 0.4 e$^-$ would translate into a potential energy difference of $\Delta V \approx 7$ eV. Comparing the work functions of TaS$_2$, $W_{TaS_2} = 5.6$ eV[22] and Au(111), $W_{Au} = 5.31$ eV[23], it is clear that potential energy differences on the order of $\Delta V \approx 7$ eV are unexpected. Indeed, estimates for the charge transfer $\Delta N$ based on work function differences[24,25] generally arrive at $\Delta N$ much less than 0.1 e$^-$ per unit cell for substrates like Au.

We argue in the following that metallic 2d materials are instead prone to pseudodoping with substrate induced changes in Fermi areas, which do not primarily relate to charge transfer but rather to hybridization in a regime which is special to physisorbed 2d materials. To arrive at this conclusion, it is insightful to analyze also the unoccupied part of the electronic structure of the adsorbed TaS$_2$, Fig. 1c. In contrast to the almost rigid downward shift of the TaS$_2$ derived bands below the $E_F$, we find strongly momentum and energy dependent changes in the electronic dispersion above the $E_F$. In particular, avoided crossings between the Au surface state and the TaS$_2$ states strongly reshape the unoccupied part of the band structure near $\Gamma$, which indicates significant hybridization between the substrate and TaS$_2$ layer.

**Scanning tunneling spectroscopy.** While ARPES confirms the apparent electron doping of the filled state bands, we utilize scanning tunneling microscopy/spectroscopy (STM/STS) to probe the empty state density of states (see Methods). Figure 2a shows the morphology of 1H-TaS$_2$ measured by STM. STS was taken at various points within the moiré unit cell of the surface, near and far away from apparent point defects. We observe a non-dispersing state at 0.4 eV, which shifts in proximity of various defects (Fig. 2b).

To interpret the experimental results, we carried out DFT calculations with a supercell of TaS$_2$ with similar size as the moiré unit cell (see Supplementary Fig. 2), where nearly the same pseudodoping effect of band shift was observed (see Supplementary Fig. 3) as that in Fig. 1c. Thus, we continuously use the structure in Fig. 1a, b for the discussion in the following. To interpret the spectra, it is insightful to compare them to DFT-simulated STS (Fig. 2c), where we calculated the DOS inside a sphere at 5.0 Å above the upmost S atoms of TaS$_2$ for different separations $h$ between the TaS$_2$ and the Au(111) substrate. As shown in Fig. 2c, for $h_\infty$ the main peak appears near 0.9 eV above the $E_F$. As $h$ decreases, the peak is split into two parts due to the increasing hybridization between the Ta-$d$ band and the Au(111) surface state. That is, the peak splitting in the simulated STS directly reflects the avoided crossing between the two aforementioned bands visible in Fig. 1c. The lower part of the split peak in the simulated STS eventually shifts to about 0.4 eV above the $E_F$ for $h = h_{eq}$, which is in line with the STS spectrum at the site without defects (Fig. 2b). Indeed, the simulated spectrum at $h_{eq}$ well accounts for the measured spectrum at the site without defects, while the calculated spectrum for the free-standing case ($h_\infty$) deviates clearly from this STS spectrum. These findings in the defect-free case represent an experimental observation of the theoretically predicted hybridization effect.

Additional information about the interaction between the TaS$_2$ and the substrate can be obtained by inspecting STS spectra

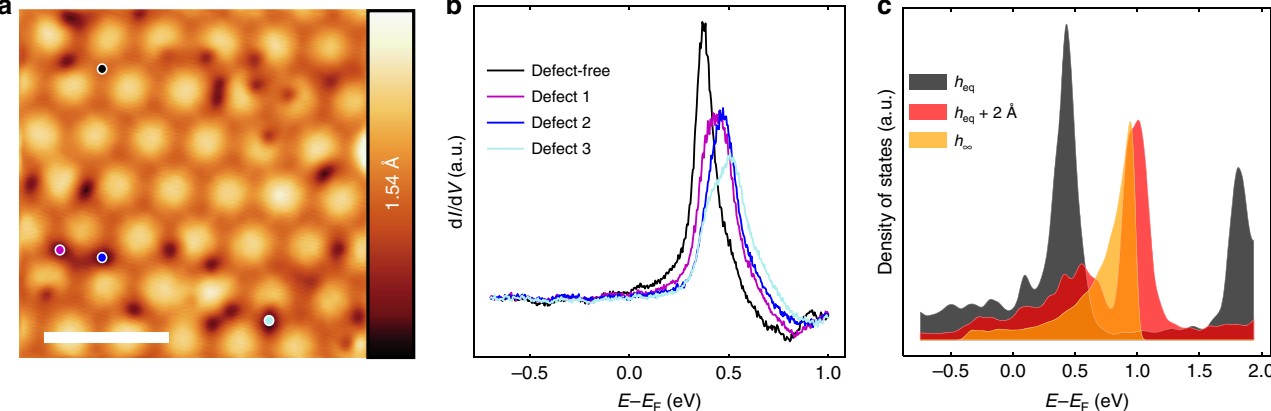

**Fig. 2** Scanning tunneling microscopy and spectroscopy of TaS$_2$ on Au(111). **a** High-resolution STM topography in the constant-current mode, where the color bar is related to the measured apparent height and the length of the scalar bar is 5 nm, STM parameters: Sample bias voltage $V_S = 335$ mV, tunneling current $I_T = 500$ pA. **b** Comparison of STS point spectra from the sites with/without defects. STS parameters: Stabilization voltage $V_{stab} = 1$ V, stabilization current $I_{stab} = 500$ pA, modulation voltage $V_{mod} = 5$ mV. The corresponding measurement sites are represented by the spots in **a**. **c** STS simulations as a function of the adsorption height $h$

collected in the vicinity of structural defects. Those spectra are also shown in Fig. 2b. The presence of the defects shifts the peak in STS spectrum to higher energies and reduces its amplitude with respect to the spectrum without defects (Fig. 2b). Interestingly, those defect-induced changes are similar to the modifications that we find in the calculated spectra, when increasing the adsorption height from $h_{eq}$ to $h_{eq} + 2$ Å (Fig. 2c).

Detailed studies of $MoS_2$ on Au(111) found spots of missing Au atoms right at the $MoS_2$/Au(111) interface[26,27], where the distance between the 2d material and the substrate is larger than on the perfect surface. For these kinds of defects it appears natural that the defect-induced changes in the measured tunneling spectra (Fig. 2b) are very similar to the simulated spectra (Fig. 2c) with increased adsorption height $h$. Still, the microscopic nature of the defects found in our experiment remains speculative and further candidates are e.g., sulfur vacancies within the $TaS_2$ layer. However, we expect that almost any defect at or close to the Au (111) surface regardless of its exact microscopic nature will influence the surface state and its coupling to the $TaS_2$ conduction band.

Taken together, the STS experiments confirm the significant impact of the hybridization on the electronic structure of the adsorbed $TaS_2$. We argue in the following that this hybridization provides the key to understanding the apparent heavy doping of the $TaS_2$ monolayer on the Au(111) substrate in such a way that the nominal occupancy of the $TaS_2$ states varies considerably, while keeping the actual amount of charge in $TaS_2$ orbitals almost the same as in the free-standing layer.

**Minimal model of pseudodoping.** We consider a minimal model involving two bands derived from the orbital $|a\rangle = (1,0)$ of the 2d material and $|b\rangle = (0,1)$ of substrate, where we assume a (real valued) hybridization $t_\perp \geq 0$ and a constant offset $-2\Delta$ in the on-site energies between the two. The resulting Hamiltonian reads

$$\hat{H}(k) = \varepsilon_0(k)\mathbf{1} + t_\perp \sigma_1 - \Delta\sigma_3, \quad (1)$$

where we used the Pauli matrices $\sigma_i$ and summarized all $k$-dependence of the initial dispersion in $\varepsilon_0(k)$. The eigenstates of the full Hamiltonian are

$$|-\rangle = \cos(\varphi)|a\rangle - \sin(\varphi)|b\rangle,$$
$$|+\rangle = \cos(\varphi)|b\rangle + \sin(\varphi)|a\rangle \quad (2)$$

with $\tan 2\varphi = t_\perp/\Delta$ and corresponding energies

$$\varepsilon_\mp(k) = \varepsilon_0(k) \mp \sqrt{\Delta^2 + t_\perp^2}. \quad (3)$$

As shown in Fig. 3a, the hybridization yields an admixture of substrate derived states to the 2d material states and vice versa, as well as a level repulsion between the hybridizing bands.

With $E_F = 0$, the ground state density matrix reads

$$\hat{N}(k) = \Theta(-\varepsilon_-(k))|-\rangle\langle-| + \Theta(-\varepsilon_+(k))|+\rangle\langle+|. \quad (4)$$

The central point is now to contrast the occupation of the bands

$$N_\mp = \int d^2k \langle\mp|\hat{N}(k)|\mp\rangle \quad (5)$$

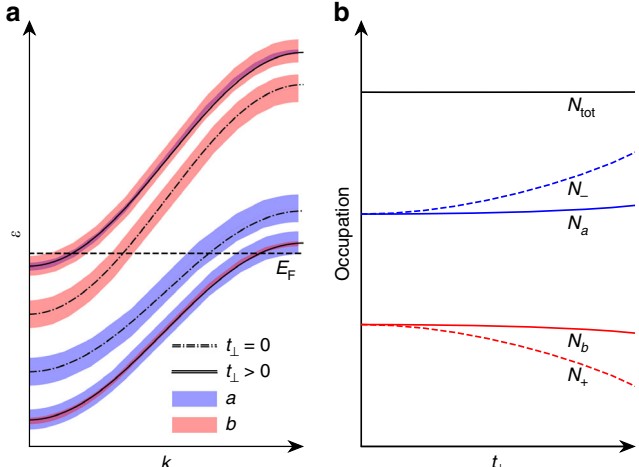

**Fig. 3** Minimal two-band model of pseudodoping. Evolution of **a** the energy dispersion and **b** the orbital/band occupation $\left(N_{a/b}/N_\mp\right)$ as a function of the hybridization ($t_\perp$) between the orbital $|a\rangle$ of 2d materials and $|b\rangle$ of substrates. As illustrated in **a**, the hybridization leads to an admixture of substrate derived states (red) to the 2d material states (blue) and a level repulsion between the hybridizing bands. The level repulsion, as $t_\perp$ increases, allows the band occupation $N_\mp$ to vary sizably, i.e., the upper band $|+\rangle$ (the lower band $|-\rangle$) is apparently doped by holes (electrons), while leaving the orbital occupation $N_{a/b}$ almost unchanged as shown in **b**. $N_{tot}$ refers to the total occupation number of the system, which is a constant under the variation in $t_\perp$

with the occupation of the orbitals

$$N_i = \int d^2k \langle i|\hat{N}(k)|i\rangle, \text{ where } i \in \{a, b\}. \quad (6)$$

The nominal band occupations $N_\mp$ are those measured in ARPES. In contrast, the orbital occupations $N_a$ and $N_b$ correspond to the actual charges in the 2d layer and the substrate, respectively, manifest in core level spectroscopies and also determine electric fields at the interface. Here, we have $N_a = \cos^2(\varphi)N_- + \sin^2(\varphi)N_+$ and $N_b = \cos^2(\varphi)N_+ + \sin^2(\varphi)N_-$.

For vanishing hybridization ($t_\perp = 0 \Rightarrow \varphi = 0$) the orbital occupancies simply coincide with the band occupancies:

$$N_{a/b}^0 = N_\mp^0 = \int_{-\infty}^0 d\varepsilon \rho^0(\varepsilon \pm \Delta),$$

where $\rho^0(\varepsilon) = \int d^2k\delta(\varepsilon - \varepsilon_0(k))$ is the DOS associated with the dispersion $\varepsilon_0(k)$. At finite $t_\perp$ the band fillings become

$$N_\mp = \int_{-\infty}^0 d\varepsilon \rho^0\left(\varepsilon \pm \sqrt{\Delta^2 + t_\perp^2}\right)$$
$$= N_\mp^0 + \int_{-\infty}^0 d\varepsilon\left[\rho^0\left(\varepsilon \pm \sqrt{\Delta^2 + t_\perp^2}\right) - \rho^0(\varepsilon \pm \Delta)\right] \quad (7)$$
$$\approx N_\mp^0 \pm \rho^0(0)t_\perp^2/2\Delta.$$

I.e. in ARPES experiments, the upper (lower) band appears hole (electron) doped by an amount of $\rho^0(0)t_\perp^2/2\Delta$ (c.f. dashed lines in Fig. 3b).

**Table 1 DFT estimates of the actual number of electrons in the Ta-$d$ orbitals $N_a$ (per Ta atom), corresponding actual charge transfer $\Delta N_a$ and apparent charge transfer $\Delta N_-$ as a function of adsorption height $h$ of the TaS$_2$ above the Au (111) surface**

| $h$ | $N_a$ | $\Delta N_a$ | band shift (eV) | $\Delta N_-$ |
|---|---|---|---|---|
| $h_{eq}$ | 3.44 | 0.04 | ~0.20 | ~0.40 |
| $h_{eq}+1$ Å | 3.41 | 0.01 | ~0.03 | ~0.06 |
| $h_{eq}+2$ Å | 3.40 | 0 | ~0.01 | ~0.02 |
| $h_{eq}+3$ Å | 3.40 | 0 | 0 | 0 |
| $h_{eq}+4$ Å | 3.40 | 0 | 0 | 0 |

However, the occupancy of the orbital $|a\rangle$ in the 2d layer is

$$
\begin{aligned}
N_a &= \cos^2(\varphi)N_- + \sin^2(\varphi)N_+ \\
&\approx N_-^0 + \rho^0(0)\left[t_\perp^2/(2\Delta) + \varphi^2(-2\Delta)\right] \\
&= N_-^0 + O(t_\perp^3).
\end{aligned}
\tag{8}
$$

Analogously, we find $N_b \approx N_+^0$ to second order in $t_\perp$ (c.f. solid lines in Fig. 3b). I.e. the actual charge transfer cancels to leading order in $t_\perp$ while the nominal doping does not. This is the key point of the pseudodoping mechanism. The hybridization, as depicted in Fig. 3, allows the occupancies $N_-$ of the lower band mainly contributed by the 2d material states to vary considerably, while keeping the actual amount of charge $N_a$ in the 2d material almost the same as in its free-standing form.

While the model outlined here is generic, we can obtain estimates of the actual charge transfer $\Delta N_a = N_a - N_a^0$ and the apparent doping $\Delta N_- = N_- - N_-^0$ for the specific case of TaS$_2$ on Au (111) as function of adsorption height from our DFT calculations: $N_a$ is obtained by integrating the Ta-$d$ projected DOS from energy $-\infty$ to the Fermi energy, which becomes a constant after increasing $h$ beyond $h_{eq}+3$ Å. Hence, the TaS$_2$ is decoupled from the Au substrate for $h$ larger than $h_{eq}+3$ Å and we correspondingly obtain $\Delta N_a(h) = N_a(h) - N_a(h_{eq}+4$ Å$)$. $\Delta N_-$ is calculated from the shift $\Delta E$ of the TaS$_2$ band at a given adsorption height, via $\Delta N_- = \Delta E \rho^0(0)$, where $\rho^0(0) \approx 2$ eV$^{-1}$ has to be the total density of states of free standing TaS$_2$ at the Fermi level. As shown in Table 1, $\Delta N_-$ is about 10 times larger than $\Delta N_a$ at the fully relaxed adsorption height ($h = h_{eq}$). When lifting the TaS$_2$ off the surface, $\Delta N_-$ and $\Delta N_a$ decrease rapidly. Both findings are well in line with the generic notion of pseudodoping.

**Pseudodoping and electronic instabilities**. Interaction terms like Coulomb interactions naturally couple localized states. In the example of a Hubbard type interaction they are of the form

$$
H_U = U \sum_i \hat{n}_{i\uparrow}^a \hat{n}_{i\downarrow}^a,
\tag{9}
$$

where $i$ refers to the lattice site of the electrons in the 2d layer, $\sigma = \{\uparrow, \downarrow\}$ to their spin and $\hat{n}_{ia}^a$ are the corresponding occupation number operators. $U$ is the on-site interaction matrix element and can describe repulsive ($U > 0$) or attractive interaction ($U < 0$).

We assume that the substrate states are non-interacting and that their DOS at the Fermi level is small as compared to the DOS of the 2d material. Then, the admixture of the substrate states to the bands of the 2d material upon hybridization reduces the effective interaction inside the hybridized band by a factor of $Z^2 = \cos^4(\phi)$, i.e., $U \to U^{\mathrm{eff}} = Z^2 U$.

For weak coupling instabilities such as BCS superconductivity, characteristic transition temperatures $T_C \sim \exp[-1/|U^{\mathrm{eff}}\rho|]$ are determined by the interaction $U^{\mathrm{eff}}$ and DOS at the $E_F$ $\rho$ resulting from the hybridized band. If the DOS of the original 2d band is

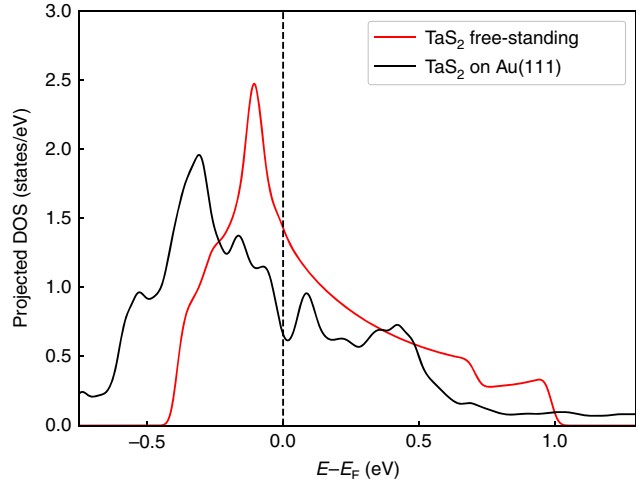

**Fig. 4** Influence of the Au(111) substrate on the TaS$_2$ DOS. Comparison of the DOS projected on Ta-$d$ orbitals of 1H-TaS$_2$ on Au(111) substrate (black line) and in its free-standing form (red line)

structureless, we expect simply a reduction of $T_C$. An analogous line of argumentation holds for weak coupling charge- or spin-density wave instabilities at some wave vector $Q$, where the susceptibility $\chi(Q) = \int d^2k(f(\varepsilon_k) - f(\varepsilon_{k+Q}))/(\varepsilon_k - \varepsilon_{k+Q})$ plays the role of an effective density of states. Thus, hybridization effects should quench tendencies towards weak coupling electronic instabilities, if the DOS/generalized susceptibilities are essentially independent of the $E_F$. An even stronger suppression of instabilities is expected if $\rho$ or $\chi(Q)$ are reduced upon hybridization related pseudodoping. A comparison of the DOS of 1H-TaS$_2$ on Au(111) and in its free-standing form (c.f. Fig. 4) shows that there is indeed a reduction of $\rho(E_F)$ upon deposition on Au, i.e., both, the reduction of effective coupling constants and the reduced DOS[28], will contribute here to the suppression of charge density wave/superconducting states observed in ref. [12].

If, however, hybridization shifts a strong peak in the DOS of the 2d material's band towards $E_F$ such that the hybridization induced increase in $\rho$ overcompensates the reduction of the effective interaction by the factor $Z^2$, we arrive at an increase in $T_C$. Additionally, if there is a sizable interaction between electrons in the substrate states, the effective interaction can indeed be enhanced upon hybridization with the substrate, which finally also yields an increase in $T_C$. Such an inherited interaction from the substrate has been conjectured to contribute to the strongly enhanced superconducting critical temperature of FeSe when coupled to SrTiO$_3$[29].

## Discussion

In summary, we have introduced the mechanism of pseudodoping, which can lead to apparent heavy doping of 2d materials on metallic substrates even in case of weak physisorption. Pseudodoping manifests itself as surprisingly large shifts of bands in ARPES experiments, as measured for TaS$_2$ on Au(111)[12]. Pseudodoping is expected to be a general phenomenon in metallic 2d materials grown on 2d metallic substrates. It is due to hybridization of metallic 2d materials with electronic states from the supporting substrates and involves much less actual charge transfer between the layer and its substrate than changes in the Fermi areas would suggest. Nonetheless, instabilities of the electronic system towards symmetry broken states are expected to be highly sensitive to this kind of doping. Particularly, the prospect of utilizing pseudodoping to control electronic phase diagrams deserves future explorations.

## Methods

**Experimental details**. Substrates were clean Au(111), prepared by repeated cycles of Ar$^+$ sputtering and annealing (at 850–970 K) until a sharply defined spin-split surface electronic state was clearly visible with ARPES. Metallic Ta was evaporated from a hot Ta filament (Goodfellow, 99.9% purity, 0.25 mm diameter) in an H$_2$S atmosphere (Linde, 99.5% purity) of approximately high $10^{-5}$ – low $10^{-4}$ mbar (base pressure low $10^{-10}$ mbar). During deposition, samples were held at elevated temperatures in the range 870–970 K. Deposition rates were between 0.5 and 1 monolayers per hour. After sample growth, the samples were kept exclusively in ultra-high vacuum (UHV) conditions, with no exposure to air. Sample transport was in a UHV transport suitcase chamber.

After transport, samples were prepared for STM/STS measurements by annealing in UHV for approximately 1 h at 970 K. The measurements were conducted with an Omicron LT-STM operated at 5 K. The base pressure was better than $1 \times 10^{-10}$ mbar. An electrochemically etched tungsten wire, flashed in situ, was used as the tip. The bias voltage $V_S$ was applied to the sample. Topography images were taken in constant-current mode. Spectroscopy was performed utilizing a lock-in technique and the differential conductance was recorded with a modulation voltage of $V_{mod} = 5$ mV at a frequency of $f_{mod} = 5356$ Hz. The STS curves shown in Fig. 2b stem from a set of in total 250 spectra. Each shown curve is averaged over 5 spectra on the same spot. Each type of defect was probed at different spots, altogether at least 30 times. The measured peak energies of one type at the same spot vary less than ±1 mV. For the same type of defects measured at different spots of the sample, i.e., in different spots of the moiré, the variation is larger (see Supplementary Table 1) but the standard deviation in the peak positions is still decisively smaller than the defect induced peak shifts.

**Computational details**. We performed density functional theory calculations using the Vienna Ab Initio Simulation Package (VASP)[30] with the projector augmented wave basis sets[31,32] and the generalized gradient approximation (GGA) to the exchange correlation potential[33]. In all cases we fixed the in-plane lattice constant of 1H-TaS$_2$ to the experimental value of $a = 3.316$ Å[34]. Calculations with the lattice constant of $a = 3.337$ Å from ref. [12] yield virtually indistinguishable results.

The Au(111) surfaces were modeled using slabs with thickness of 30 atomic layers and a single layer of 1H-TaS$_2$ absorbed on the upper side of the slabs (see Fig. 1a, b in the main text). The Au(111) slab was furthermore terminated with H atoms on the bottom side of the slab, i.e., on the side without TaS$_2$ coverage. In the direction perpendicular to the slab, a vacuum separation of more than 14 Å was used to avoid the interaction between the successive surfaces, which was reduced when varying the adsorption height. I.e., the size of the unit cell was fixed in all calculations. The lateral coordinates of all atoms were kept fixed and we laterally compressed the Au(111) surface by 0.5% to match a $(\sqrt{3} \times \sqrt{3})$ R30° unit cell of 1H-TaS$_2$ with a $2 \times 2$ unit cell of the metal surface. The vertical coordinates of the Ta and S atoms were relaxed until forces acting on them were below 0.01 eV Å$^{-1}$ leading to structures, where the closest vertical distance between S atoms and Au surface atoms is 2.37 Å. A Γ-centered $8 \times 8 \times 1$ $k$-point mesh and a plane wave cut-off energy of 350 eV were used in structural relaxations, which were increased to $21 \times 21 \times 1$ and 400 eV in the electronic structure calculations, respectively. The self-consistent criterion of energy convergence is $10^{-4}$ eV. We have accounted for van der Waals interaction according to the D2 method of Grimme[35] in all the calculations. The van der Waals interactions are decisive in determining the adsorption height of the TaS$_2$ layer: GGA calculations without van der Waals interaction yield an adsorption height of 2.86 Å, which is 0.5 Å more than in the case with van der Waals interactions. Correspondingly, pseudodoping induced band shifts would be about 0.1 eV smaller in the case without van der Waals interactions.

We checked the influence of the stacking registry between the TaS$_2$ $(\sqrt{3} \times \sqrt{3})$ R30° supercell and the Au(111) $2 \times 2$ supercell on the pseudodoping by considering the three geometries shown in Supplementary Fig. 1 and found virtually indistinguishable upfolded band structures in all cases (see Supplementary Note 1). We furthermore considered a $7 \times 7$ supercell of TaS$_2$ on an $8 \times 8$ supercell of Au (111) (Supplementary Fig. 2) which corresponds to the moiré unit cell observed in STM. Also for this geometry we find pseudodoping and hybridization effects which are very similar to the case of the TaS$_2$ $(\sqrt{3} \times \sqrt{3})$ R30° supercell and the Au(111) $2 \times 2$ supercell (see Supplementary Fig. 3 and Supplementary Note 2).

## Data availability

The data from this work can be obtained from the corresponding author upon reasonable request.

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

## Acknowledgements

This work was supported by the European Graphene Flagship and DFG RTG 2247, by the Danish Council for Independent Research, Natural Sciences under the Sapere Aude program (Grant No. DFF-4002-00029) and by VILLUM FONDEN via the Centre of Excellence for Dirac Materials (Grant No. 11744). We acknowledge financial support from the European Research Council ERC Grant Agreement No. 339813 (Exchange) and The Netherlands Organization for Scientific Research (NWO) via the VIDI project: 'Manipulating the interplay between superconductivity and chiral magnetism at the single atom level' with project number 680-47-534. The numerical computations were carried out on the Norddeutscher Verbund zur Förderung des Hoch- und Höchstleistungsrechnens (HLRN) cluster.

## Author contributions

B.S. and T.O.W. performed and analyzed the DFT calculations. C.S., A.S.N., M.B. and P.H. fabricated the samples; A.E., C.S., A.S.N., and A.A.K. performed the STM measurements. All authors contributed to the discussion and analysis of the results and the writing of the manuscript.

## Additional information

**Competing interests:** The authors declare no competing interests.

**Journol peer review information**: *Nature Communications* thanks the anonymous reviewers for their contribution to the peer review of this work. Peer reviewer reports are available.

