## [Peer Review File · Nature Communications]

Reviewers' comments:

Reviewer #1 (Remarks to the Author):

The manuscript submitted by Shao and co-authors reports an interesting and comprehensive investigation of the interaction of a 2H-TaS₂ monolayer (ML) and Au(111) substrates. It combines theoretical work (DFT+model hamiltonian) and STS experiments, to propose a mechanism of « pseudodoping » of 2d materials ML on metallic substrate. This mechanism is proposed to explain the presence of large band shifts in ARPES experiments, already reported in Ref [12] and is based on the hybridization of metallic 2d materials orbitals with surface electronic states of the substrate. The key issue is that this process involves much less actual charge transfer between the two subsystems than the changes in Fermi areas suggest. This manuscript presents new informations as well as new hypotheses of work on the promising study of metallic ML based on Transition Metal Dichalcogenide (TMDC). I have found the present communication interesting for a broad audience, and the data partially convincing. However I would like to invite the authors to revise their work to address the specific concerns listed below:

— The computational details section are by far too scarce to allow for a colleague to reproduce any DFT calculation presented here. No crucial informations like plane-wave cut-off energy, k-points sampling, nor vacuum height are provided for instance.

— Why H atoms were adsorbed on the opposite side of the Au slab ? Usually to avoid spurious dipolar moment, corrections can be added...

— The use of PBE xc-functional to describe such weakly bound systems, in which physisorption is supposedly the key interaction is questionable. Since no experimental evidence of the substrate-ML distance is available, what happens if one tries a van der Waals-corrected xc-functional ?

— The authors only report electronic structures calculations for one adsorption geometry. I do think like for graphene on fcc (111) surface, especially when hybridization is present like in PRB 71,075402, several models should be addressed.

— What are the effects of Spin-Orbit Coupling on the band structures ?
It is known that for many TMDCs, SOC strongly influence the band dispersion.

— Maybe the potential readers could take benefit of a detailed discussion of the d-type orbitals involved in the hybridization of the Ta atoms with Au the p_z orbitals. It could also be a chance to help to refine the proposed hamiltonian model ?

— Could the authors compare their DFT values of work function with the ones used in the discussion on page 2 ? To my opinion, using bulk work function values is misleading, since from bulk to ML, work function usually change significantly.

— to support the discussion based on the model hamiltonian, I strongly suggest the authors to corroborate the behavior of Fig.3(b) with DFT estimates, extracted from integrated projected DOS for instance with respect to the ML-substrate distance. In the same spirit, several charge density integration schemes could be used to effectively prove that there are only weak charge transfer.

— More importantly the use of pseudodoping word is not entirely convincing to me. I don't see the novelty/difference with what it is called hybridization of orbitals between two subsystems... In this sense I strongly suggest the authors to revise the title.

Reviewer #2 (Remarks to the Author):

The manuscript by Bin Shao et al. reports a combination of experiments and ab-initio calculations on the electronic properties of 1H-TaS₂ metallic monolayers grown onto Au(111) surface. Authors interest is on the so called "pseudodoping" mechanism, consisting in considerable changes of Fermi areas (of the 2D material) that arise from weak hybridization of the 2D material and the supporting substrate, even in case of physisorption. As authors claim such mechanism to be ubiquitous for metallic 2D materials grown onto metals, they explore the TaS₂/Au case as a representative example. Authors explicitly use a previous study (Ref.12) - reporting ARPES and STM/STS data on the same system TaS₂/Au - and integrate such background knowledge by means of new STM/STS data and theory.

The manuscript is well written and clear and the figures support the authors' claims. I report below some issues that authors should clarify:

- at pag. 3, authors should mention Fig. 2(b) when introducing for the first time discussion on the STS spectra (e.g. "[...] We observe a non-dispersing state at 0.4 eV, which shifts in proximity of various defects (Fig. 2(b))").

- In the caption of Fig. 2, which is the meaning of V_{stab} and I_{stab} ? Do they refer to the starting tunnelling conditions when acquiring the STS spectra?

- How many raw STS spectra have been averaged to obtain the curves reported in Fig. 2(b)? Which is the typical dispersion (noise) of spectra acquired at the same site? I wonder if differences between the three defects are robust against statistical fluctuations.

- Did the authors find evidences, in the STS spectra acquired far or close to point defects, of the second peak predicted by DFT-simulated STS to be roughly at or above 1.0eV? From discussion it seems that such second peak would represent a key signature supporting the suggested hybridization picture described by DFT.

- Also, it seems that authors are thinking to specific defects, that do not significantly alter the electronic surface state of Au. I wonder if this is plausible? Could authors comment further, for example with some supporting reference?

- abstract reports "1T-TaS₂" but in the main text "1H-TaS₂" or simply TaS₂ are reported. Please remove ambiguity.

Reviewer #3 (Remarks to the Author):

The proposed model for pseudo-doping of two-dimensional materials is interesting and probably valuable, nevertheless the presented data and calculation need further refining.

The authors use a lattice constant (0.3316nm) dating back to 1973(ref 34) while only two years ago some among this group proposed a different lattice constant(0.3337) for their study on the same system. The difference is small but now the TaS₂($\sqrt{3}$)X($\sqrt{3}$)R30 become 5.78 and the Au(111)2X2 is 5.76. Therefore it does not justify a compressive stress applied on the on the calculated Au surface. Moreover the STM image show a low resolution moiree pattern with a periodicity just below 2.5nm. Suggesting that perhaps the measured moiree is closed to a Au(111)8x8 in coincidence with perhaps TaS₂(7x7) or TaS₂(4 $\sqrt{3}$ X 4 $\sqrt{3}$)R30.

Therefore it is not granted that the model used for the calculation is really representative of what experimentally measured.

It is my humble opinion that the calculations should be performed on the same structure that meant to represent: the structure measured in STM.

Point-by-point reply to Reviewer 1:

Comment: The computational details section are by far too scarce to allow for a colleague to reproduce any DFT calculation presented here. No crucial information like plane-wave cut-off energy, k-points sampling, nor vacuum height are provided for instance.

Response: We agree with the reviewer and correspondingly provide all computational details required for unambiguous reproducibility in the method section of the revised manuscript.

Comment: Why H atoms were adsorbed on the opposite side of the Au slab? Usually to avoid spurious dipolar moment, corrections can be added...

Response: Since the surface state wave functions of the Au surface can extend deeply into the material, we adsorbed H atoms on the bottom of the slab to suppress the surface state there and thus to avoid the interaction between the surface state on the top and bottom of our slab model. That is not so decisive in the calculations with substrate thicknesses of 30 Au layers but very important in the calculations with slab thickness of 5 Au layers.

Comment: The use of PBE xc-functional to describe such weakly bound systems, in which physisorption is supposedly the key interaction is questionable. Since no experimental evidence of the substrate-ML distance is available, what happens if one tries a van der Waals-corrected xc-functional?

Response: We thank the referee for this very important point: We have performed additional calculations including structural relaxations, where we have accounted for van der Waals interactions according to the D2 method of Grimme [S. Grimme, J. Comp. Chem. 27, 1787 (2006)]. All relaxed structures discussed in the revised manuscript are now with van der Waals (vdW) interactions taken into account. The generic effect of the vdW interaction is that the adsorption height h_{eq} of the TaS₂ layer is decreased by about 0.5 Å. As a consequence, there is stronger hybridization between TaS₂ and Au(111) resulting in stronger pseudodoping: the downward shift of the TaS₂ conduction band increases from 0.1 eV in the case without vdW correction to 0.2 eV in the case with vdW correction. We correspondingly updated the main text and state explicitly in the revised methods section the importance of the van der Waals interaction in determining the strength of pseudodoping.

Comment: The authors only report electronic structure calculations for one adsorption geometry. I do think like for graphene on fcc (111) surface, especially when hybridization is present like in PRB 71,075402, several models should be addressed.

Response: In order to address this well-chosen point we performed a series of additional calculations, where we have considered three possible stacking arrangements for the slab model of TaS₂($\sqrt{3}\times\sqrt{3}$)R30 on the Au(111) 2x2 surface, as shown in Supplementary Fig.1. We find very similar adsorption heights (differing only by < 0.02 Å) in all cases. The resulting upfolded band structures highlighting the bands with large Ta-d character are compared in the lower panel of Supplementary Fig. 1. The occupied parts of the upfolded band structures are virtually indistinguishable and there is hardly any influence of the exact stacking registry on the band shifts and hybridization effects. I.e. the pseudodoping effect is essentially independent of the exact stacking registry chosen in the supercell calculations. Indeed, the major quantity determining the strength of the coupling between the TaS₂ and the Au(111) substrate states is the adsorption height. Since the lowest energy stacking sequence belongs to geometry type "C" of Supplementary Fig.1, all data we show in the main text of the revised manuscript also correspond to this geometry now (instead of type "A" shown formerly). As the electronic structure is essentially independent of the exact adsorption registry, there are no changes regarding the interpretations / conclusions we draw. We added an explanation regarding the robustness of our results with respect to different stacking geometries on page 2 of the revised manuscript.

Comment: What are the effects of Spin-Orbit Coupling on the band structures? It is known that for many TMDCs, SOC strongly influence the band dispersion.

Response: We have performed additional calculations which account for spin-orbit coupling (SOC). The figure below shows an unfolded band structure including SOC with the Ta-*d* derived states highlighted in blue (computational details see caption). Most importantly: the shift of the bands is -0.2 eV as in the case without SOC, i.e. SOC splits the TaS₂ bands but does not affect the band shifts significantly. The major parameter that really strongly influences the “pseudodoping” strength is the adsorption height as explained above.

Reply Fig. 1: Comparison of unfolded band structures including spin-orbit coupling of TaS₂ on Au(111) with free standing TaS₂. We considered a monolayer of TaS₂ ($\sqrt{3}\times\sqrt{3}$)R30° on the Au(111) 2x2 surface with a thickness of 5 atomic layers and stacking sequence of type “C”. The bottom side of the Au slab was terminated by H atoms. A vacuum separation of more than 6.5 Å in the direction perpendicular to the slab was used to avoid the interaction between the successive surfaces. Otherwise the same computational parameters as detailed in the methods section have been used.

Comment: Maybe the potential readers could take benefit of a detailed discussion of the d-type orbitals involved in the hybridization of the Ta atoms with Au the p_z orbitals. It could also be a chance to help to refine the proposed hamiltonian model ?

Response: There are three different orbital characters involved in TaS₂ conduction band. Strong hybridization between TaS₂ and electronic states from the Au(111) substrate is particularly obvious for Ta dz². However, the situation in the real material is more complicated than in the model Hamiltonian since the latter does not involve the exact orbital character of the 2d materials band. Lifting the model Hamiltonian to e.g. a full three band tight-binding description of TaS₂ to fit the exact TaS₂ band structure and also to account for all orbital character would be technically possible. However, the model gets very involved and less generic, then, and it will be also very difficult then to arrive at a simple and intuitive analytical explanation of the pseudodoping effect, which is indeed a generic one. We believe that the combination of full DFT calculations accounting for all atomistic details for our example system and a simplistic generic, easily analytically understandable model as currently done in the paper is most insightful. In order to still, strengthen the link between the generic model and the DFT calculations we think it is very helpful to follow the after next suggestion by the referee on the DFT estimates of different kinds of charge transfers.

Comment: Could the authors compare their DFT values of work function with the ones used in the discussion on page 2 ? To my opinion, using bulk work function values is misleading, since from bulk to ML, work function usually change significantly.

Response: In our DFT calculations of the free standing monolayer we find a work function for TaS₂ of $W=5.92$ eV which is slightly larger than the experimental value of $W=5.6$ eV measured for the surface of bulk TaS₂. Still the work function difference with Au (111) ($W_{Au}=5.31$ eV) is very small as compared to the potential energy difference of $\Delta W= 7$ eV associated with a direct charge transfer of $0.4e^-$ from Au (111) to TaS₂. In the former version of the manuscript where we neglected the van der Waals interaction we were discussing a work function difference on the order of 4eV due to the smaller “pseudodoping” we found there. Regardless of whether or not we account for the van der Waals interaction and regardless of whether we account for the (experimental) bulk or (calculated) monolayer work function of TaS₂ it is clear that work function differences cannot explain the “doping” we find here.

Comment: to support the discussion based on the model hamiltonian, I strongly suggest the authors to corroborate the behavior of Fig.3(b) with DFT estimates, extracted from integrated projected DOS for instance with respect to the ML-substrate distance. In the same spirit, several charge density integration schemes could be used to effectively prove that there are only weak charge transfer.

Response: We would to thank the referee for her/his very good idea and follow her/his suggestion by adding Table 1 to the main text, where we exactly compare DFT estimates of the apparent doping, “ ΔN ”, as measured in e.g. ARPES and the actual charge transfer “ ΔN_a ” as function of adsorption height. “ N_a ”, i.e. the actual charge in the Ta-d atomic spheres is simply obtained by integrating the Ta-d projected DOS from energy $-\infty$ to the Fermi energy. “ ΔN ”, i.e. the apparent doping measured in ARPES, is obtained by multiplying the shift of the TaS₂ band at a given adsorption height with the total density of states of free standing TaS₂ at the Fermi level. We find that “ ΔN ” exceeds “ ΔN_a ” by a factor of roughly 10 at the fully relaxed equilibrium distance of TaS₂ on Au(111). “ ΔN ” and “ ΔN_a ” decrease rapidly when lifting TaS₂ off the Au(111) surface. Both findings are well in line with the simplistic pseudodoping model depicted in Fig. 3. A corresponding discussion of Table 1 has been added on page 8 of the main text.

Comment: More importantly the use of pseudodoping word is not entirely convincing to me. I don't see the novelty/difference with what it is called hybridization of orbitals between two subsystems... In this sense I strongly suggest the authors to revise the title.

Response: Hybridization is a very generic concept, which is always effective whenever two quantum mechanical systems are in direct contact. Correspondingly, there is a wide range of phenomena ranging from molecular bonding to Kondo and heavy fermion physics which is related to hybridization between different orbitals. Thus, we feel that the term hybridization is not specific enough to characterize concisely what we observe here: we essentially find “doping as manifested in changes of Fermi areas without actual charge transfer”. This phenomenon is a *consequence* of hybridization, while hybridization is much more unspecific. The term “pseudodoping” is chosen to condense the idea of “doping without actual charge transfer” in a single word. As such “pseudodoping” refers to the specific consequence, while “hybridization” would refer to the very generic cause. This is why we think that it is appropriate to have the word “pseudodoping” in the title and to explain that pseudodoping is due to hybridization in the abstract.

Point-by-point reply to Reviewer 2:

Comment: at pag. 3, authors should mention Fig. 2(b) when introducing for the first time discussion on the STS spectra (e.g. "[...] We observe a non-dispersing state at 0.4 eV, which shifts in proximity of various defects (Fig. 2(b))").

Response: We thank the referee for this point. Indeed, we should mention Fig. 2b in the sentence suggested by the referee. We added that correspondingly in the revised manuscript.

Comment: In the caption of Fig. 2, which is the meaning of V_{stab} and I_{stab} ? Do they refer to the starting tunnelling conditions when acquiring the STS spectra?

Response: Yes, as suspected by the referee "stab" stands indeed for "stabilization", i.e. the voltage and tunneling current the tip was stabilized at before starting the measurement. To clarify the meaning of "stab" we explicitly define " V_{stab} " and " I_{stab} " now in the caption of Fig. 2.

Comment: How many raw STS spectra have been averaged to obtain the curves reported in Fig. 2(b)? Which is the typical dispersion (noise) of spectra acquired at the same site? I wonder if differences between the three defects are robust against statistical fluctuations.

Response: The curves shown in Fig. 2b stem from a set of in total 250 spectra. Each shown curve is averaged over 5 spectra on the same spot. Each type of defect was probed at different spots, altogether at least 30 times. The measured peak positions of one type at the same spot vary less than ± 1 mV. For the same type of defects measured at different spots of the sample, i.e. in different spots of the moiré, the variation is somewhat larger (see table below) but the standard deviation in the peak positions is still decisively smaller than the defect induced peak shifts. Hence, we conclude that the measured defect induced peak shifts are indeed robust against statistical fluctuations. To demonstrate this robustness also to the readers of the paper we added the table below also to the supplemental material and refer to it in the methods section of the revised main text.

Type	Avg. peak energy	Std. dev.	Number of spots
No defect	373 mV	± 6 mV	20
Defect 1	431 mV	± 6 mV	7
Defect 2	455 mV	± 10 mV	11
Defect 3a	411 mV	± 17 mV	6
Defect 3b	512 mV	± 17 mV	6

Reply table 1: Summary of statistical fluctuations in STS experiments. Each defect has been measured at as many spots as indicated in the column "number of spots" and for each spot we averaged over 5 spectra, i.e. in total $5 \cdot (20+7+11+6+6)=250$ measurements. The resulting average peak energies and standard deviations in the peak energies are given in the second and third column, respectively.

Comment: Did the authors find evidences, in the STS spectra acquired far or close to point defects, of the second peak predicted by DFT-simulated STS to be roughly at or above 1.0eV? From discussion it seems that such second peak would represent a key signature supporting the suggested hybridization picture described by DFT.

Response: Unfortunately, we do not have STS data above +1.0eV. However, we think that this does not hinder the conclusion we draw on hybridization effects above the Fermi level. Indeed, the presence of the very pronounced peaks in our STS around 0.4eV is only explicable if hybridization of the TaS₂ conduction band with the Au (111) surface state is taken into account: A peak around 0.4eV is absent in the DFT

simulations of free standing TaS₂, in stark contrast to our experimental STS and the simulated STS which include the hybridization.

Comment: Also, it seems that authors are thinking to specific defects, that do not significantly alter the electronic surface state of Au. I wonder if this is plausible? Could authors comment further, for example with some supporting reference?

Response: We *do* expect that the defects influence the electronic surface state of Au. The defects very likely shift the surface state in energy as even noble gas adsorbates like Xe do. (See e.g. Ref. 21 / PRB 78, 161408 (R) (2008) for an ARPES study of adsorbate induced surface state shifts.) Defects at the interface of the related system of MoS₂ on Au (111) have been studied in Ref. 26 (Nano Lett. 16, 5163 (2016)) and one type of common defects there were “pits” of missing Au atoms in the upmost Au (111) surface layer. For these defects it appears very natural to have STS spectra which resemble locally the effect of changing adsorption heights. Still, the microscopic nature of the defects remains speculative and other candidates include e.g. sulfur vacancies. However, we expect that almost any defect at or close to the Au(111) surface regardless of its exact microscopic nature (i.e. regardless whether it is e.g. a missing S-atom in the TaS₂ or a missing Au atom in the Au(111) surface) will influence the surface state. As a consequence, also any defect will change the coupling of the Au (111) surface state to the TaS₂ conduction band and thus likely also affect (e.g. shift) resonance around +0.4eV in our STS, which originates from the coupling between the Au (111) surface to the TaS₂ conduction band. That is indeed what we observe in the vicinity of the defects regardless of their microscopic nature. We think that the formulation regarding the influence of the defects on the Au surface state was not clear enough in the previous version and we thus amended it in the revised manuscript on page 3.

Comment: abstract reports "1T-TaS₂" but in the main text "1H-TaS₂" or simply TaS₂ are reported. Please remove ambiguity.

Response: We thank the referee for this point. Indeed, there has been a misprint in the abstract of the former version: It has to be 1H-TaS₂. We corrected the abstract correspondingly.

Point-by-point reply to Reviewer 3:

Comment: The authors use a lattice constant (0.3316nm) dating back to 1973(ref 34) while only two years ago some among this group proposed a different lattice constant(0.3337nm) for their study on the same system. The difference is small but now the TaS₂($\sqrt{3}\times\sqrt{3}$)R30 become 5.78 and the Au(111)2X2 is 5.76. Therefore it does not justify a compressive stress applied on the on the calculated Au surface.

Response: We checked to which extent the change in TaS₂ lattice constant from 0.3316nm to 0.3337nm has an influence on the pseudodoping effect by performing additional calculations with lattice constant 0.3337nm. The resulting upfolded band structures are shown in the figure below. The difference between the upfolded band structures is completely insignificant and we thus keep the data with 0.3316nm in the manuscript. We add a comment to the manuscript that working with a lattice constant of 0.3337nm does not change any of our results in the methods section of the main text on page 6.

Reply Fig. 2: Comparison of upfolded band structures of TaS₂ on Au(111) with different TaS₂ lattice constants of 0.33316nm (left) and 0.3337nm (right). In both cases, we considered a monolayer of ($\sqrt{3}\times\sqrt{3}$)R30° on Au(111) 2x2 slab with a thickness of 30 atomic layers, stacking sequence of type “C and otherwise the same computational parameters as detailed in the methods section of the main text.

Comment: Moreover the STM image show a low resolution moiree pattern with a periodicity just below 2.5nm. Suggesting that perhaps the measured moiree is closed to a Au(111)8x8 in coincidence with perhaps TaS₂(7x7) or TaS₂(4 $\sqrt{3}\times\sqrt{3}$)R30. Therefore it is not granted that the model used for the calculation is really representative of what experimentally measured. It is my humble opinion that the calculations should be performed on the same structure that meant to represent: the structure measured in STM.

Response: We performed additional calculations using exactly the supercell model (see Supplementary Fig. 2) suggested by the referee which fits indeed the experimentally measured moiré very well. A comparison of the upfolded band structure resulting from TaS₂(7x7) on an 8x8 supercell of Au(111) to the upfolded band structures for TaS₂($\sqrt{3}\times\sqrt{3}$)R30 on 2x2 Au (111) is given in Supplementary Fig. 3 and also below.

Regardless of the lateral supercell size we find similar changes in the TaS₂ upfolded dispersion upon adsorption on Au (111): there is always a hybridization induced plateau around Γ and a downward shift of the occupied states on the order of -0.2eV. The major difference between the different supercell models arises from the different slab thicknesses. In the case of “only” 5 Au layers quantum confinement leads to rather coarse quantization of the vertical crystal momentum k_z of the bulk Au conduction bands. Coupling of these k_z quantized Au bands to the TaS₂ conduction band leads to a few avoided crossing with apparent opening of minigaps in the TaS₂ which goes over into a broadening of the TaS₂ band in the case of thicker slabs (see model with 30 atomic Au layers) as it should be. Hence, we conclude that the thickness of the Au slab is the more critical parameter than the lateral extent of the supercell model. That is also in line with

the finding of pseudodoping being mainly independent of the exact stacking realized at the interface of TaS₂ and Au(111), as can be seen from Supplementary Fig. 1. To make this point clear we added Supplementary Fig. 2 & 3 in the revised version and discuss in the main text on page 2 and in the methods section the influence of exact stacking and moiré superlattice on the upfolded band structures.

Reply Fig.3: (Same as Supplementary Fig. 3) Comparison of upfolded band structures (blue dots) from differently sized supercells to the DFT dispersion of free standing TaS₂ (undoped (black) and shifted by 0.2eV (red)).

REVIEWERS' COMMENTS:

Reviewer #1 (Remarks to the Author):

Considering that all the comments/questions suggested by the referees have been carefully addressed and the manuscript's quality has been thus clearly improved, I do recommend it for publication.

Sincerely yours

Reviewer #2 (Remarks to the Author):

The authors have satisfyingly addressed my original comments in the revised version of the manuscript.

Reviewer #3 (Remarks to the Author):

Only two remarks.

The paragraph "Discussion" would be better if named "Conclusions".

As stated in such paragraph this behaviour is expected to be ubiquitous, and they prove it to be applicable to one system.

The general statement of the title should therefore be reconsidered, since as it is now it appears as overbroadening/misleading the contents of the experimental evidences presented.

Point-by-point reply to Reviewer 3:

Comment: The paragraph "Discussion" would be better if named "Conclusions".

Response: Since the heading "Discussion" is requested by *Nature Communications*, we kept it in the final version of the manuscript.

Comment: As stated in such paragraph this behaviour is expected to be ubiquitous, and they prove it to be applicable to one system.

The general statement of the title should therefore be reconsidered, since as it is now it appears as overbroadening/misleading the contents of the experimental evidences presented.

Response: While the experimental data is indeed for one system, the theoretical model is very general and expected to be applicable to other 2d metallic materials. In our opinion it is justified to make general statements based on theory. How to exactly reflect the situation here in the title and in the paper is of course a matter of taste. We think that changing the title to "Pseudodoping of a Metallic Two-Dimensional Material by The Supporting Substrate" and replacing the word "ubiquitous" in the Discussion section by "a general phenomenon" is most appropriate and we changed the paper accordingly.